# Delayed Spiking Neural Network and Exponential Time Dependent Plasticity Algorithm

## Abstract

Spiking Neural Networks (SNNs) become more similar to artificial neural networks (ANNs) to solve complex machine learning tasks. However, such similarity does not bring superior performances but loses biological plausibility. Moreover, most learning methods of SNNs follow the pattern of gradient descent used in ANNs, which also suffer from low bio-plausibility. To address these issues, a realistic delayed spiking neural network (DSNN) is introduced in this study, which only considers the dendrite and axon delays as the learnable parameters. And a more biologically plausible exponential time-dependent plasticity (ETDP) algorithm is proposed to train the DSNN. The ETDP adjusts the delays according to the global and local time differences between presynaptic and postsynaptic spikes, and the forward and backward propagation time of signals. These biological indicators can surrogate the time-consuming computation of descents precisely. Experimental results demonstrate that the DSNN trained by ETDP achieves very competitive results on various benchmark datasets, compared with other SNNs.

## 1 Introduction

Recently, spiking neural networks (SNNs) have attracted more and more attention, due to their event-driven property, biological plausibility and neuromorphic hardware realization Rao et al. (2022). Plenty of SNNs have been proposed including Hodgkin-Huxley model, FitzHugh-Nagumo model FitzHugh (1961), Morris-Lecar model Morris & Lecar (1981), and Integrate-and-Fire (IF) model Gerstner & Kistler (2002). For simplification, the IF and its leaky version (LIF) are the most commonly used models to construct SNNs. The main obstacle to SNNs is the lack of effective learning algorithms. Spike-timing dependent plasticity (STDP), as a specific version of Hebbian learning, is one of the most popular unsupervised learning for SNNs Caporale & Dan (2008). But the performance of SNN trained by STDP is limited on complicated machine learning tasks, without supervised signals.

More SNNs appeal to the supervised learning algorithms based on gradient descent (GD), which are commonly used to train ANNs. More directly, ANN-to-SNN conversion approaches have been proposed to convert the trained ANNs into rate-coded SNNs. That improves the efficiency of SNNs on neuromorphic hardware Rueckauer et al. (2016); Sengupta et al. (2019); Fang et al. (2021). However, such conversion only enables the computation capability of SNNs to approximate that of ANNs, but never exceed. What is worse, these approaches will greatly degenerate the biological plausibility of SNN, which is the most characteristic of SNNs distinguished from ANNs.

Biological evidence has proven the existence of delays in the mammalian neocortex, which can be modulated according to input and output spikes Madadi Asl et al. (2017). In addition to synaptic weights, delays are usually used as the auxiliary learnable parameters to improve the model capabilities in SNNs Yu et al. (2022); Luo et al. (2022). However, such approaches do not fully consider the biologically plausible property of delays in natural nervous systems, since delays can directly affect the precise timing of spikes. Therefore, a delayed spiking neural network (DSNN) and an exponential time dependent plasticity (ETDP) algorithm are proposed in this study. Both the DSNN and ETDP are equipped with high levels of biological plausibilities, which are different from most conventional SNNs. Comprehensive experiments are conducted to evaluate the performance of proposed approaches. To sum up, the main contributions of this study can be described as follows:

- A delay-based DSNN is proposed, using only dendrite and axon delays as learnable paramters instead of elaborate parameters in other SNNs.

- A biologically plausible ETDP is introduced to train the DSNN. In ETDP, the global and local time dependences between presynaptic and postsynaptic spikes decide whether the dendrite and axon delays increase or decrease, and the forward and backward propagation time of spiking signals determine the change amplitudes of delays.

- Compared with other SNNs and learning methods, the DSNN trained by ETDP achieved satisfactory performances on machine learning tasks and retained high levels of biological plausibility simultaneously.

## 2 RELATED WORKS

SpikeProp is one of the earliest SNN models that temporally encode information in terms of spike timing Bohte et al. (2002). It formulates the spike timing as a function of the neuron's membrane potential and a GD-based learning algorithm to minimize the timing differences between output and desired spikes. Later, several improved versions have been proposed, such as Extending SpikeProp Schrauwen & Van Campenhout (2004), Multispike SpikeProp Booij & tat Nguyen (2005), Quick-Prop and Rprop McKennoch et al. (2006). Another SNN with a temporal coding scheme is proposed in Mostafa (2017). By deducing a brief formulation of the relation between input and output spike times, MSNN works very similarly to conventional ANNs. It can be trained on large-scale datasets with the aid of GPU acceleration. Following research that attempts to improve the MSNN can be found in Comsa et al. (2020); Zhou et al. (2021); Fang et al. (2021).

Tempotron is a biologically plausible supervised learning algorithm to train spike timing-based neurons Gütig & Sompolinsky (2006). It enables a single LIF neuron to encode categorical feature information in the latencies of single spikes or synchronized multi-spikes instead of spike counts. Afterward, an extended multi-spike tempotron named aggregate-label learning is introduced to train LIF neuron models on the temporal credit assignment problem Gütig (2016). It improves the capacity of neurons by producing multiple output spikes, rather than simply giving out binary spiking and non-spiking signals. In Yu et al. (2018), a threshold-driven plasticity algorithm is proposed to improve the efficiency of multi-spike tempotron, based on the linear assumption of threshold. Recently, Qin et al. (2023) proposed an attention-based temptron, in which the output spikes are divided into clusters. Then, these spike clusters are used to encode information instead of discrete spikes. The spike timing-dependent plasticity (STDP) is used to train the specific SNN, whose two-layer framework consists of different kinds fo neurons, the layer of excitatory neurons and the other layer of inhibitory ones Diehl & Cook (2015). It achieves very satisfactory performance on the image recognition problem in an unsupervised learning way.

In general, there are two main training methods for SNNs, including the GD-based supervised learning algorithm and the STDP-based unsupervised learning algorithm. The GD algorithm is commonly used to train conventional ANNs. It is accurate and efficient to optimize the object function. However, it seems impossible that such a specific error calculated by derivation can be implemented in the brain at the same time. Compared with the GD algorithm, the STDP algorithm is more biologically plausible. It has been reported in the nervous system of various species. But, its optimization performance is relatively limited as an unsupervised learning algorithm.

Naturally, plenty of research attempts to gain advantages from both GD and STDP methods and avoid their shortcomings simultaneously. For instance, ReSuMe is a supervised learning method that enables the SNN to code neural information in precise spike timing trains Ponulak & Kasiński (2010). It employs the interaction between two STDP processes, based on well-recognized physiological phenomena. SPAN applies the well-known Widrow–Hoff rule to adjust the synaptic weights, and achieves a desired input/output spike behavior in a supervised fashion Mohemmed et al. (2012). To train multilayer SNNs, a nonlinear voltage-based learning rule named SuperSpike is proposed in Shrestha & Orchard (2018). It enables the deterministic IF neurons to perform nonlinear computations on spatiotemporal spike patterns. SLAYER is a new backpropagation mechanism to train synaptic weights and axonal delays, which uses a temporal credit assignment policy to backpropagate the error to each preceding layer Shrestha & Orchard (2018). It can overcome the non-differentiable problem of spike functions, and train the SNNs with deep architectures. In addition,

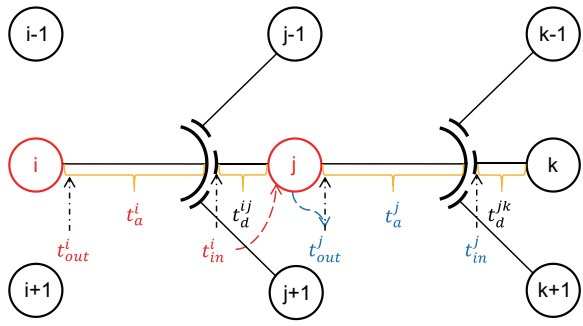

Figure 1: Delayed spiking neural network with three fully-connected layers. $i$, $j$ and $k$ are the indexes of neurons on the input, hidden and output layers, respectively.

other surrogate gradient learning algorithms can be found in Neftci et al. (2019); Zhang & Li (2020); Zheng et al. (2021).

## 3 DELAYED SPIKING NEURAL NETWORK

The architecture of DSNN with three fully-connected layers has been presented in Fig. 1. A modified IF neuron with an exponentially decaying synaptic current kernel is used as the computing unit of DSNN. The kernel merely considers the time delays of signals passing through the axons, synapses and dendrites, to replace the synaptic weights which are commonly used in the conventional SNNs. The kernel function ($\kappa$) of IF neurons can be described by

$$\kappa(t - t_{in}^i) = \Theta(t - t_{in}^i)e^{(-\frac{t - t_{in}^i - t_d^{ij}}{\tau})},$$
$$with \quad \Theta(t) = \begin{cases} 1 & \text{if } t \geqslant 0 \\ 0 & \text{otherwise,} \end{cases} \tag{1}$$

where $t_{in}^i$ represents the time of spike from neuron $i$ arriving at the postsynaptic structure of the next neuron $j$. $\tau$ denotes a time constant that determines the speed of the exponential decaying of spike-caused synapse current. $t_d^{ij}$ is the dendrite delay, modeling the dendrite structure's time effect in integrating input spikes and then causing the membrane potential change of neuron $j$. As illustrated in Fig. 1, $t_{out}^i$ is the firing time of neuron $i$, $t_a^i$ represents the axon delay which contains the time of spike traveling through the axon of neuron $i$, and the time of electrical-chemical signal transduction within the presynaptic structure. Thus, we can get $t_{in}^i = t_{out}^i + t_a^i$. It is notable that, once a neuron fires, it will not be permitted to produce any spikes again in the same trial.

Each neuron of DSNN will receives $N$ spikes from the previous neurons in a trial at times $\{t_{in}^1, t_{in}^2, \cdots, t_{in}^N\}$. As presented in Fig. 2, the membrane dynamic of neurons can be described by

$$\frac{du^j(t)}{dt} = \sum_{i=1}^{N} \kappa(t - t_{in}^i), \tag{2}$$
$$with \ B.C. \ T_{i-1}(t_{in}^i) = T_i(t_{in}^i),$$

where $u^j$ is the membrane potential of neuron $j$. The given boundary conditions ($B.C.$) are based on the assumption of the membrane potential's continuity. And $T_k$ is given by

$$T_k(t) = \sum_{i=1}^{k} \tau(-e^{-\frac{t - t_{in}^i - t_d^{ij}}{\tau}}) + C_k, \tag{3}$$

where $k \in \{1, 2, \cdots, N\}$, $C_k$ is the integration constant. By solving Eq. 2 (details in Appendix B), for $t < t_{out}^j$, the membrane potential of neuron $j$ is given by

$$u^j(t) = \sum_{i=1}^{N} \Theta(t - t_{in}^i) \, e^{\frac{t_d^{ij}}{\tau}} \cdot \tau(1 - e^{-\frac{t - t_{in}^i}{\tau}}). \tag{4}$$

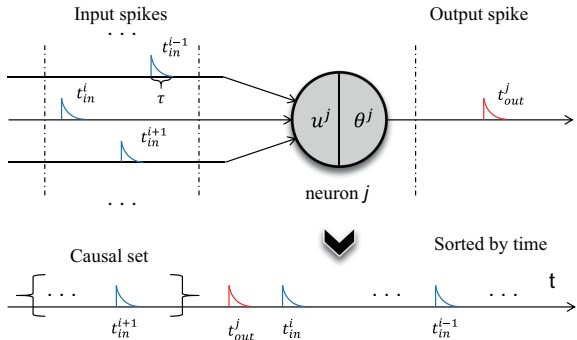

Figure 2: Causal set of a single neuron in the DSNN.

Considering neuron $j$ spikes at the output time $t_{out}^j$, only the input spikes that arrived before $t_{out}^j$ can influence the output time. The subset of these spikes is termed as the causal set of input spikes $C = \{i | t_{in}^i < t_{out}^j\}$, as illustrated in Fig. 2. Let $\theta^j$ denote the firing threshold of neuron $j$, we can get $u^j(t_{out}^j) = \theta^j$. Then, Eq. 4 will be transformed into

$$\theta^j = \sum_{i \in C} e^{\frac{t_d^{ij}}{\tau}} \cdot \tau (1 - e^{-\frac{t_{out}^j - t_{in}^i}{\tau}}). \tag{5}$$

Then, $t_{out}^j$ can be implicitly defined by

$$t_{out}^j = \tau \ln\left(\frac{\sum_{i \in C} e^{\frac{t_d^{ij}}{\tau}} e^{\frac{t_{in}^i}{\tau}}}{\sum_{i \in C} e^{\frac{t_d^{ij}}{\tau}} - \frac{\theta^j}{\tau}}\right). \tag{6}$$

For the simplification, the time constant $\tau$ is set to 1, and $e^{t_x}$ is definied as $z_x$. Finally, Eq. 6 can be transformed into

$$z_{out}^j = \frac{\sum_{i \in C} z_d^{ij} z_{in}^i}{\sum_{i \in C} z_d^{ij} - \theta^j}. \tag{7}$$

This transformation of variables yields a direct expression relating input spike times to the output spike times in the exponential time domain. In Eq. 7, it is easy to observe that the sum of the exponential dendrite delays of the causal input spikes should be larger than the threshold; otherwise, they could not cause the neuron to fire. Algorithm 2 presentes pseudocode of the forward process of DSNN in Appendix A.5.

## 4 EXPONENTIAL TIME DEPENDENT PLASTICITY

As introduced above, there are two learnable parameters in the DSNN, namely the dendrite delay $z_d$ and the axon delay $z_a$. First, the conventional gradient descent (GD) algorithm is applied as the training algorithm, by minimizing the mean square error function between the desired targets $z_T$ given by the supervised signals and the final outputs of the entire network $z_O$. Take a DSNN with three fully connected layers as an example in Fig. 1, whose layers contain $I$, $J$ and $K$ neurons, respectively. In the framework of the GD algorithm, the loss function of the $K$ output neurons can be presented as

$$E = \frac{1}{2} \sum_{k=1}^{K} (z_O^k - z_T^k)^2. \tag{8}$$

According to Eq. 7, $z_O^k$ is computed as follows,

$$\begin{aligned} z_{in2}^j &= z_{out1}^j \cdot z_{a1}^j = z_{a1}^j \frac{\sum_{i \in C^1} z_{d1}^{ij} \cdot z_{in1}^i}{\sum_{i \in C^1} z_{d1}^{ij} - \theta_1^j}, \\ z_O^k &= z_{out2}^k \cdot z_{a2}^k = z_{a2}^k \frac{\sum_{j \in C^2} z_{d2}^{jk} \cdot z_{in2}^j}{\sum_{j \in C^2} z_{d2}^{jk} - \theta_2^k}, \end{aligned} \tag{9}$$

where $z_{in1}$ and $z_{out1}$ are the input and output of the hidden layer, and $z_{in2}$ and $z_{out2}$ are those of the output layer. $z_{d1}$, $z_{a1}$, $z_{d2}$ and $z_{a2}$ are the corresponding dendrite and axon delays of these two layers. The partial derivatives of $E$ with respect to $z_{d2}$ and $z_{a2}$ of the output layer are

$$\frac{\partial E}{\partial z_{d2}^{jk}} = \frac{\partial E}{\partial z_O^k} \frac{\partial z_O^k}{\partial z_{d2}^{jk}} = \begin{cases} (z_O^k - z_T^k) \cdot \frac{z_{a2}^k(z_{in2}^j - z_{out2}^k)}{\sum_{p \in C^2} z_{d2}^{pk} - \theta_2^k}, & j \in C^2; \\ 0, & otherwise. \end{cases} \quad (10)$$

$$\frac{\partial E}{\partial z_{a2}^k} = \frac{\partial E}{\partial z_O^k} \frac{\partial z_O^k}{\partial z_{a2}^k} = (z_O^k - z_T^k) \cdot z_{out2}^k. \quad (11)$$

Defining $A = \frac{1}{\sum_{p \in C^2} z_{d2}^{pk} - \theta_2^k}$, it can keep positive if the neuron fires as introduced above. The dynamic of the parameters $z_{d2}^{jk}$ and $z_{a2}^k$ can be written as

$$\begin{aligned} \dot{z}_{d2}^{jk} &= A \cdot R_{d2}^k \cdot (z_O^k - z_T^k) \cdot (z_{in2}^j - z_{out2}^k), j \in C^2, \\ \dot{z}_{a2}^k &= F^k \cdot (z_O^k - z_T^k), \end{aligned} \quad (12)$$

where $R_{d2}^k = z_{a2}^k$ called the reverse timing term, which denotes the exponential timing of error signal passing **reverse** to the dendrite between neurons $k$ and $j$. And $F^k = z_{out2}^k$ is called the forward timing term, which denotes that of input signal passing **forward** to the neuron $k$. The portion $z_O^k - z_T^k = e^{t_O^k} - e^{t_T^k}$ is named the global timing difference term, whose direction is decided by the contrast of the final output time $t_O^k$ and the target time $t_T^k$. And the other portion $z_{in1}^i - z_{out1}^j = e^{t_{in1}} - e^{t_{out1}}$ is named the local timing difference term, whose direction is determined by the input and output times of neurons in the current layer. Similarly, the dynamic of the parameters $z_{d1}$, $z_{a1}$ of the hidden layer can be presented as follows:

$$\begin{aligned} \dot{z}_{d1}^{ij} &= A' \cdot R_{d1}^{ij} \cdot (z_O^k - z_T^k) \cdot (z_{in1}^i - z_{out1}^j), j \in C^2 \wedge i \in C^1, \\ \dot{z}_{a1}^j &= B' \cdot R_{a1}^j \cdot F^j \cdot (z_O^k - z_T^k), j \in C^2, \end{aligned} \quad (13)$$

where $A' = \frac{1}{\sum_{p \in C^2} z_{d2}^{pk} - \theta_2^k} \cdot \frac{1}{\sum_{q \in C^1} z_{d1}^{qj} - \theta_1^j}$, and $B' = \frac{1}{\sum_{p \in C^2} z_{d2}^{pk} - \theta_2^k}$. Both $A'$ and $B'$ are constant positive during the backpropagation process. The reverse timing term $R_{d1}^{jk} = z_{a2}^k \cdot z_{d2}^{jk} \cdot z_{a1}^j$, denotes the exponential timing of error signal passing **reverse** to the dendrite between neurons $i$ and $j$. The other reverse timing term $R_{a1}^{jk} = z_{a2}^k \cdot z_{d2}^{jk}$, denotes the reverse exponential timing to the axon of neuron $j$. The forward timing term $F^j = z_{out1}^j$ represents the exponential timing of input signal passing **forward** through the neuron $j$.

Eqs. 12 and 13 imply that the sign of global term determines the increase or decrease of the axon delays, and the signs of global and local terms decide whether the dendrite delays increase or decrease. The forward and reverse timing terms control the change amplitudes of the dendrite and axon delays. By initializing large dendrite delays or setting small threshold values, the terms $A$, $A'$ and $B'$ are approximately equal to small positive constant. With the learning rates of $\eta_d$ and $\eta_a$, the update of dendrite and axon delay between neuron $i$ and $j$ can be given by

$$\begin{aligned} \Delta z_{d2}^{jk} &= -\eta_d \cdot R_{d2}^k \cdot (z_O^k - z_T^k) \cdot (z_{in2}^j - z_{out2}^k), j \in C^2, \\ \Delta z_{a2}^k &= -\eta_a \cdot F^k \cdot (z_O^k - z_T^k), \\ \Delta z_{d1}^{ij} &= -\eta_d \cdot R_{d1}^{jk} \cdot (z_O^k - z_T^k) \cdot (z_{in1}^i - z_{out1}^j), j \in C^2 \wedge i \in C^1, \\ \Delta z_{a1}^j &= -\eta_a \cdot R_{a1}^{jk} \cdot F^j \cdot (z_O^k - z_T^k), j \in C^2. \end{aligned} \quad (14)$$

It is interesting to find that, both the global and local timing difference terms are similar to the classic STDP algorithm. In the STDP, whether to strengthen or weaken the weights is dependent on the signs of time differences $t_{in} - t_{out}$ between the pre-synaptic spikes $t_{in}$ and post-synaptic spikes $t_{out}$ Legenstein et al. (2005); Andrade-Talavera et al. (2023). However, the exponential terms of time differences $e^{t_{in}} - e^{t_{out}}$ in the Eqs. 12 and 13 can approximate the theoretic gradients more accurately. Thus, our learning algorithm is named exponential time dependent plasticity (ETDP). It is worth noting that these foundations can be easily generalized to the DSNN with multiple hidden layers in a deep architecture, by simply updating the forward and reverse timing terms.

---

**Algorithm 1:** Backward process of ETDP in one layer of the DSNN

---

**Input:** Causal set of neurons in this layer $C$, global term of the EDTP $\delta = z_O - z_T$, number of neurons in the previous and current layers $M, N$, learning rates for dendrite and axon delays $\eta_d, \eta_a$;

**Output:** Updated dendrite and axon delays $z_d, z_a$;

1 **for** $i = 1$ *to* $M$ **do**
2     **for** $j = 1$ *to* $N$ **do**
3        **if** $j \notin C$ **then**
4           $\delta^{'}[i]$ += $0$;
5           $\Delta z_d[i, j]$ += $0$;
6           $\Delta z_a[j]$ += $0$;
7        **else**
8           $\delta^{'}[i]$ += $\delta[i] \cdot z_d[i, j] \cdot z_a[j]$;
9           $\Delta z_d[i, j]$ += $\eta_d \cdot \delta[j] \cdot z_a[j] \cdot (z_{out}[j] - z_{in}[i])$;
10           $\Delta z_a[j]$ += $\eta_a \cdot \delta[j] \cdot (-z_{out})$;
11        **end**
12     **end**
13 **end**
14 $z_d \leftarrow z_d + \Delta z_d$;
15 $z_a \leftarrow z_a + \Delta z_a$;
16 **return** $z_d$ and $z_a$.

---

## 5    BIOLOGICAL PLAUSIBILITY ANALYSIS

Neuroscientific studies have conclusively demonstrated the ubiquity of spiking time delay in the mammalian neocortex, underscoring its pivotal role in the signal processing of the nervous system. For instance, the dynamic regulation of synaptic latency at neocortical and hippocampal excitatory synapses has been experimentally validated in Boudkkazi et al. (2011); Rama et al. (2015). Egger et al. observed that conduction delays along intracortical axons strongly influence neural activity patterns Egger et al. (2020). Madadi et al. highlighted the significant impact of dendritic and axonal propagation delays on the STDP algorithm in determining the neural structures of SNNs Madadi Asl et al. (2017).

Recognizing the indispensability of spiking time delays, various studies have sought to incorporate them into SNNs. For instance, synaptic delay is integrated with synaptic weights to enhance the learning ability of ReSuMe on spiking neurons Taherkhani et al. (2015), and a modified multispike learning approach is proposed, which jointly considers synaptic weight and delay Yu et al. (2022). Both dendritic and axonal delays have been adopted to augment the computational capability of spiking neural P systems Garcia et al. (2021). Additionally, a reconfigurable axon delay is incorporated into hardware realizations of SNNs to improve performance Ochs et al. (2021).

However, in the aforementioned works, spiking time delays are typically treated as additional learnable parameters alongside synaptic weights, resulting in increased complexity and longer running times for the learning methods Taherkhani et al. (2020). In contrast, our proposed DSNN uniquely employs only dendritic and axon delays as learnable parameters. Notably, the exponential term of dendritic delays in our model ($e^{t_d^{ij}}$) is functionally analogous to the synaptic weights ($w_{ij}$) used in conventional IF neurons, if we rewrite Eq. 2 as follows:

$$\frac{du^j(t)}{dt} = \sum_{i=1}^{N} e^{\frac{t_d^{ij}}{\tau}} \cdot \Theta(t - t_{in}^i) e^{(-\frac{t - t_{in}^i}{\tau})}. \tag{15}$$

The primary motivation behind our study is to emphasize the significance of time delays in the signal processing of SNNs, aligning with neurobiological observations. Our experiments illustrate that equipping SNNs solely with dendritic and axon delays, without the commonly used synaptic weights, can still yield competitive performance. To the best of our knowledge, our DSNN is the first neuron network that exclusively relies on dendritic and axon delays as learnable parameters.

As emphasized in Taherkhani et al. (2020), delay is an inherent property of real biological systems, and incorporating a proper delay learning mechanism can enhance the biological plausibility of learning algorithms and improve the processing ability of SNNs. Consequently, we introduce a novel ETDP to train the time delays of DSNN. Eq. 14 reveals that the strengthening or weakening of delays is determined by the global and local time differences between presynaptic and postsynaptic spikes, and the change amplitude is related to the forward and backward propagation time of signals. For simplicity, the update strategy of ETDP can be presented as follows:

$$\Delta z^{jk} = \eta \cdot (z_{out}^j - z_{in}^k) \cdot (z_O - z_T) \cdot (R^k \cdot F^k). \tag{16}$$

Notably, this learning algorithm, originally inspired by the gradient-based approach, intriguingly bears a striking resemblance to the biologically plausible learning rule outlined in Roelfsema & Holtmaat (2018). The formulation is as follows:

$$\Delta w^{jk} = \eta \cdot H(t_{in}, t_{out}) \cdot RPE(t_O, t_T) \cdot FB_k, \tag{17}$$

where $\Delta w^{jk}$ denotes the change of connection strength between neurons $j$ and $k$. The function $H(\cdot, \cdot)$ represents the formalized version of Hebb's rule, with the presynaptic activity ($t_{in}$) and postsynaptic activity ($t_{out}$). The term $RPE(\cdot, \cdot)$ corresponds to the reward-prediction error function, quantifying differences between the obtained and expected rewards for the network's output ($t_O$ and $t_T$). Neurophysiological studies have revealed that the RPE signal can be induced by in vivo neuromodulatory systems. Specifically, the dopaminergic system produces a positive RPE to strengthen the tagged synapses, while the adenosine system generates a negative RPE to diminish synaptic strength Izhikevich (2007); Frémaux & Gerstner (2016); Fisher et al. (2017). In addition, it is worth noting that the $RPE$ is a global signal that influences all the neurons in the network.

The term $FB_k$ represents feedback from higher brain regions onto neuron $k$, originating from the motor and frontal cortex areas of the brain Moore & Armstrong (2003); Jonikaitis & Deubel (2011). Importantly, $FB$ remains positive and varies between 0 and 1. Thus, it gates the synaptic plasticity but never changes the sign. As described above, it becomes evident that the learning rule presented in Eq. 17 operates in a manner highly similar to our proposed ETDP algorithm. This observation strongly supports the assertion that ETDP can be considered a biologically plausible learning algorithm.

## 6 Experiments

To evaluate effectiveness of the DSNN and ETDP algorithm, three benchmark classification problems are used in the experiments. The axon and dendrite delays are initialized with the random values of Gaussian distribution in the appropriate zone.

### 6.1 XOR Problem

The XOR problem is the simplest nonlinear classification problem. It is used to verify whether the neural network can solve linearly inseparable problems. In this task, the input layer of DSNN receives two spike signals with different spike timings. Considering the DSNN encodes the neural information in terms of exponential spike timings, the early spike is set to $e^{0.01}$, and the late one is set to $e^{0.99}$. The number of neurons in the hidden layer is set to 8. The firing threshold $\theta$ is 1. And the learning rates $\eta_d$ and $\eta_a$ use a self-adapting manner ranging from 0.01 to 0.001.

With random initial parameters, the experiment is repeated for 1000 times to investigate the robustness of the model on the XOR problem. Each training iteration involved 200 trials, each of which contains four input patterns and the corresponding output pattern. Since all the parameters in our model own their specific physical representations, we do not employ extra normalization that will weaken the biological plausibility of our model. Experimental result shows that the DSNN trained by the ETDP algorithm achieved a success rate of **100%** in all the 1000 repetition experiments, implying the DSNN have a satisfactory and robust capability to solve the nonlinear problem.

### 6.2 Iris Dataset

The Iris is the first used benchmark dataset used in the experiment Fisher (1936). The features are encoded by the arrival time of input spike trains in the range [0,1]. For convenience of calculations,

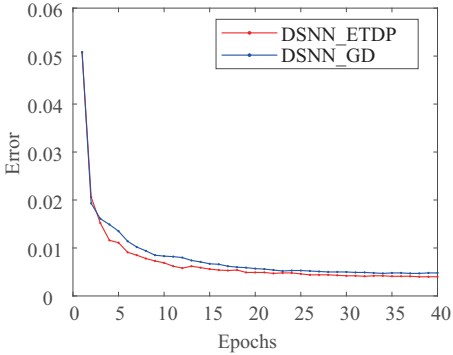

Figure 3: Convergence curves of the DSNN trained by both GD and ETDP algorithms on the MNIST dataset.

Table 1: Accuracy of SNNs on the Iris dataset.

| Models | Learning method | No. epochs | Acc (%) |
|---|---|---|---|
| SpikeProp Bohte et al. (2002) | Spiking GD | 1000 | 96.1 |
| Tempotron Gütig (2016) | Aggregate-label Learning, GD | 100 | 92.6 |
| MSNN Mostafa (2017) | Spiking GD | 400 | 96.3 |
| ReSuMe Ponulak & Kasiński (2010) | Widrow-Hoff, STDP | 200 | 94.0 |
| DL-ReSuMe Taherkhani et al. (2015) | STDP | 100 | 95.7 |
| SWAT Wade et al. (2010) | STDP | 500 | 95.3 |
| DSNN | ETDP | 400 | 96.4 |
| DSNN | GD | 400 | **96.7** |

all the spike timings are directly used in the exponential form. The neuron number of the hidden layer is set to 10. The learning rate is set to 0.1 for both axon and dendrite delays. The performance of DSNN is evaluated using 5-fold cross-validation. Compared with other SNNs, the results have been presented in Table. 1. The DSNN trained by the ETDP algorithm achieves an accuracy of **96.7%**, which is higher than all the other SNNs. That verifies the DSNN trained by the ETDP can be regarded as a useful machine learning algorithm. In addition, we also compared the performances of DSNNs trained by the ETDP and GD algorithms, respectively. The ETDP is slightly worse than the GD algorithm. This is because the ETDP only used approximate values of gradients, leading to instability during the optimization process.

### 6.3 MNIST DATASET

To assess the performance of proposed approaches on image recognition problems, the MNIST and FashionMNIST datasets are used in our experiment LeCun et al. (1998); Xiao et al. (2017). The MNIST database contains 70000 greyscale images of handwritten digits, where 60000 labeled images are used for training, and the remaining 10000 are for testing. The number of neurons is set to 800 on the hidden layer. The training epoch number is 40, with exponentially decaying learning rates ranging from $10^{-4}$ to $10^{-6}$ for both dendrite and axon delays. And the mini-batch size is set to 128. In the initialization, the values of pixels larger than 128 generate a spike at time 0.01; otherwise, produce a spike at time $ln(10) = 2.30$. We noticed that using a large temporal separation between high and low intensities will lead to better accuracy. If the separation is less than the synaptic time constant $\tau$, the boundary between input spikes would be ambiguous, which will cause accuracy degeneration. But, a too-large separation cannot further improve accuracy. It will reduce efficiency and weaken biological plausibility at the same time. As shown in Table 2, the DSNN trained by the ETDP algorithm obtains a success rate of **96.6%**, which is higher than most of the other SNNs. And the training performance of ETDP is slightly worse than that of GD. The convergence curves

Table 2: Accuracy of SNNs on the MNIST dataset.

| Models | Learning method | Acc (%) |
|---|---|---|
| MSNNMostafa (2017) | Spiking GD | 97.2 |
| Deep SNN O'Connor & Welling (2016) | Spiking GD | 96.4 |
| SG-SNN Neftci et al. (2019) | Surrogate Gradient, BPTT | 98.3 |
| Unsupervised SNN Diehl & Cook (2015) | STDP | 95.0 |
| SD-CNN Kheradpisheh et al. (2018) | STDP, SVM | **98.4** |
| Deep SCNNLee et al. (2018) | STDP, Spiking GD | 91.1 |
| DSNN | Spiking ETDP | 96.6 |
| DSNN | GD | 97.2 |

Table 3: Accuracy of SNNs on the FASHIONMNIST dataset.

| Models | Learning method | Acc (%) |
|---|---|---|
| MSNNMostafa (2017) | Spiking GD | 85.3 |
| Tempotron Gütig (2016) | Aggregate-label Learning, GD | 10.1 |
| Unsupervised SNN Diehl & Cook (2015) | STDP | 77.3 |
| DSNN | Spiking ETDP | 83.1 |
| DSNN | GD | **85.6** |

of ETDP and GD have been presented in Fig. 3, implying that the training performance of ETDP is similar to the GD algorithm again.

## 6.4 FASHIONSMNIST DATASET

In addition to the MNIST, the performance of the proposed model is also evaluated on the FASH-IONMNIST dataset, which is a more complex classification task for machine learning methods. The same hyperparameters are set as described above. As shown in Table 3, the DSNN trained by the ETDP achieves an accuracy of **83.1%**, which is a little lower than the accuracy of 85.6% of the DSNN trained by the GD algorithm, but still comparable. According to the above experiments, it can be concluded that using only the parameters of axon and dendrite delays does not affect the model capability but largely improves the biological plausibility of the DSNN. Also, the ETDP retains the optimization capability from the GD algorithm and the biological plausibility from the STDP algorithm, simultaneously.

## 7 CONCLUSIONS

Inspired by the highlights of delays in biological systems, the DSNN and ETDP learning algorithm are proposed in this study. Compared with state-of-the-art SNNs, they achieve powerful performances on various benchmark tasks, and meanwhile retain high levels of biological plausibilities. To the best of our knowledge, the DSNN is the sole spiking network architecture that only considers the dendrite and axon delays as the learnable parameters, and the ETDP is the first training algorithm that uses time differences and propagation time to substitute the gradients precisely. In our future works, we attempt to expand the DSNN with large-scale achitectures, and further simplify the computation of the ETDP algorithm without sacrificing efficiency.

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
