# A  APPENDIX

## A.1  SYNAPTIC CURRENT KERNEL OF THE DSNN

The neurons of DSNN consider the dendrite and axon delays during the information process. Specifically, the synaptic current of original IF model decays in terms of $\Theta(t - t_{in}^i)exp(-\frac{t-t_{in}}{\tau})$, while that of our modified model reduces in terms of $\Theta(t - t_{in}^i)exp(-\frac{t-t_{in}-t_d}{\tau})$. The functions of these two models have been presented in Fig. 4. It can be found that $t_d$ leads to the change of curvature for different neurons in the DSNN, rather than simply shifting the functions. In addition, a larger $t_d$ corresponds to a greater initialized synaptic current of the modified model. The foundation is in accord with biological evidence that distal synaptic inputs obtain larger local response amplitudes than similar ones at proximal locations Gulledge et al. (2005); Grillo et al. (2018). That also implies high levels of biological plausibilities in the DSNN.

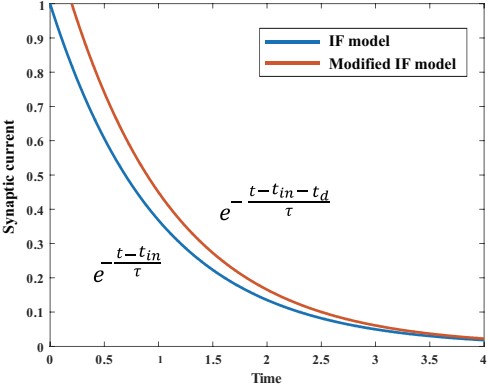

Figure 4: Comparison of synaptic currents between the original IF model and the modified one.

## A.2  SPIKING NEURAL NONLINEARITY

Artificial neural networks have been successfully applied to various fields. One of the key elements is to use nonlinear activation functions, such as the ReLU and sigmoid function. We theoretically analyze the nonlinearity of neural activation in the DSNN.

According to Eq. 7, it is easy to observe a linear dependency among the input and output spikes in terms of exponential time, assuming all input spikes are in the causal set. For example, a neuron has two input spike times $t_{in}^1$ and $t_{in}^2$, and one output spike time $t_{out}$. Let $t_{in}^1 < t_{in}^2$, there are two possible causal sets: $\{t_{in}^1\}$ and $\{t_{in}^1, t_{in}^2\}$. Their corresponding outputs can be calculated as follows:

$$\begin{cases} e^{t_{out}^\alpha} = \frac{e^{t_d^1} \cdot e^{t_{in}^1}}{e^{t_d^1} - \theta}, & C_\alpha = \{t_{in}^1\}; \\ e^{t_{out}^\beta} = \frac{e^{t_d^1} \cdot e^{t_{in}^1} + e^{t_d^2} \cdot e^{t_{in}^2}}{e^{t_d^1} + e^{t_d^2} - \theta}, & C_\beta = \{t_{in}^1, t_{in}^2\}. \end{cases} \tag{18}$$

The conditions of $C_\alpha$ and $C_\beta$ are $t_{in}^1 < t_{out}^\alpha < t_{in}^2$ and $t_{in}^1 < t_{in}^2 < t_{out}^\alpha$, respectively. By incorporating Eq. 18, we can get

$$C = \begin{cases} C_\alpha, & t_{in}^1 - t_{in}^2 \leqslant ln\frac{e^{t_d^1} - \theta}{e^{t_d^1}}; \\ C_\beta, & otherwise. \end{cases} \tag{19}$$

For certain $e^{t_d^1}$ and $\theta$ in a trial, the causal set $C$ is merely determined by the time interval $\Delta t_{in}$ between these two input spikes $t_{in}^1$ and $t_{in}^2$, where $\Delta t_{in} = t_{in}^1 - t_{in}^2$. Fig. 5(a) illustrates the function of $t_{out}$ with its two variables $t_{in}^1$ and $t_{in}^2$.

For clarity, Fig. 5(b) shows the dynamic of $t_{out}$ regarding $\Delta t_{in}$ with the fixed values of $t_{in}^1$. They are typical piecewise linear functions. During the specific ranges, a linear relationship exists between exponential input and output spike times. However, once the set of causal input spikes changes, apparent nonlinearities can be found among the interval and output. In other words, the spiking

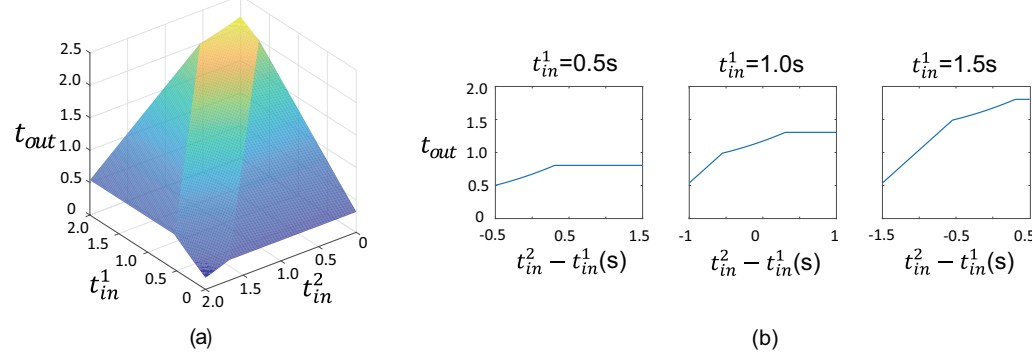

(a)                                                          (b)

Figure 5: (a) Dynamic of the output spike timing $t_{out}$ regarding the input spike timings $t_{in}^1$ and $t_{in}^2$. (b) Dynamics of $t_{out}$ regarding the time interval between input spike timings $t_{in}^2 - t_{in}^1$ with the fixed values of $t_{in}^1$.

neuron's nonlinear behavior is determined by the causal set. Such nonlinearities enable the DSNN to solve complex data mining tasks in terms of deep architectures. Besides, similar foundations can be generalized to the neuron with multiple input spikes, where the input number $N > 2$. For sorted multiple input spikes, the condition of the causal set is given by

$$C = \begin{cases} \{t_{in}^1, t_{in}^2, \cdots, t_{in}^{k-1}\}, & t_{out}^{k-1} \leqslant t_{in}^k; \\ \{t_{in}^1, t_{in}^2, \cdots, t_{in}^{k-1}, t_{in}^k\}, & t_{out}^{k-1} > t_{in}^k. \end{cases} \tag{20}$$

Considering $t_{out}^{k-1} = \frac{\sum_{i=1}^{k-1} e^{t_d^i} e^{t_{in}^i}}{\sum_{i=1}^{k-1} e^{t_d^i} - \theta}$, the conditions of $t_{in}^k$ can be transformed into

$$\begin{cases} t_{in}^k \notin C, & \sum_{i=1}^{k-1} e^{t_d^i} \cdot e^{t_{in}^i - t_{in}^k} \leqslant \sum_{i=1}^{k-1} e^{t_d^i} - \theta; \\ t_{in}^k \in C, & otherwise. \end{cases} \tag{21}$$

For the given $t_d^i$ and $\theta$, whether the $k$th spike is in the causal set is only determined by the time intervals between it and the previous spikes.

### A.3 PROOF OF THE MEMBRANE POTENTIAL OF NEURONS IN THE DSNN

More detailed proof of Eq. 6 have been presented as follows. Eq. 2 can be rewritten as

$$\frac{du^j(t)}{dt} = \begin{cases} 0, & t < t_{in}^1; \\ e^{\frac{t_d^{1j}}{\tau}} \cdot e^{-\frac{t - t_{in}^1}{\tau}}, & t_{in}^1 \leqslant t < t_{in}^2; \\ \cdots \\ \sum_{i=1}^k e^{\frac{t_d^{ij}}{\tau}} \cdot e^{-\frac{t - t_{in}^i}{\tau}}, & t_{in}^k \leqslant t < t_{in}^{k+1}; \\ \cdots \\ \sum_{i=1}^N e^{\frac{t_d^{ij}}{\tau}} \cdot e^{-\frac{t - t_{in}^i}{\tau}}, & t \geqslant t_{in}^{Nj}. \end{cases} \tag{22}$$

By integrating it, we can get

$$u^j(t) = \begin{cases} C_0, & t \leqslant t_{in}^1; \\ e^{\frac{t_d^{1j}}{\tau}} \cdot \tau(-e^{-\frac{t - t_{in}^1}{\tau}}) + C_1; & t_{in}^{1j} \leqslant t < t_{in}^{2j}. \\ \cdots \\ \sum_{i=1}^k e^{\frac{t_d^{ij}}{\tau}} \cdot \tau(-e^{-\frac{t - t_{in}^i}{\tau}}) + C_k, & t_{in}^{kj} \leqslant t < t_{in}^{k+1j}; \\ \cdots \\ \sum_{i=1}^N e^{\frac{t_d^{ij}}{\tau}} \cdot \tau(-e^{-\frac{t - t_{in}^i}{\tau}}) + C_N, & t \geqslant t_{in}^{Nj}. \end{cases} \tag{23}$$

where $C_k$ are the integration constant determined by the $B.C.$ in Eq. 2, and $k \in \{1, 2, \cdots, N\}$. Define $T_k(t) = \sum_{i=1}^{k} e^{\frac{t_d^{ij}}{\tau}} \cdot \tau(-e^{-\frac{t-t_{in}^i}{\tau}}) + C_k$, and $T_0(t) = C_0$, we can get $T_0(t_{in}^1) = T_1(t_{in}^1)$ by assuming the continuity of $u^j(t)$. With the setting $C_0 = u_{rest} = 0$, it can be written as follows:

$$e^{\frac{t_d^{1j}}{\tau}} \cdot \tau \cdot (-1) + C_1 = 0. \tag{24}$$

With the assumption $C_1 = \tau \cdot e^{\frac{t_d^{1j}}{\tau}}$ and $C_k = \tau \cdot \sum_{i=1}^{k} e^{\frac{t_d^{ij}}{\tau}}$, the condition of $B.C.$ $T_k(t_{in}^{k+1}) = T_{k+1}(t_{in}^{k+1})$ needs to satisfy the following equation:

$$\sum_{i=1}^{k} e^{\frac{t_d^{ij}}{\tau}} \cdot \tau(-e^{-\frac{t_{in}^{k+1}-t_{in}^i}{\tau}}) + \sum_{i=1}^{k} e^{\frac{t_d^{ij}}{\tau}} \cdot \tau = \sum_{i=1}^{k+1} e^{\frac{t_d^{ij}}{\tau}} \cdot \tau(-e^{-\frac{t_{in}^{k+1}-t_{in}^i}{\tau}}) + C_{k+1}. \tag{25}$$

Then, we can get $C_{k+1} = \tau \cdot \sum_{i=1}^{k+1} e^{\frac{t_d^{ij}}{\tau}}$. According to the mathematical induction, we can finally proof that $C_0 = 0$, and $C_k = \tau \cdot \sum_{u=1}^{k} e^{\frac{t_d^{uj}}{\tau}}$, where $(k = \{1, 2, \cdots, N\})$. Thus, $T_k(t)$ can be rewrited as:

$$T_k(t) = \tau \cdot \sum_{i=1}^{k} e^{\frac{t_d^{ij}}{\tau}} (1 - e^{-\frac{t-t_{in}^i}{\tau}}). \tag{26}$$

Collecting all the $T_k(t)$, the membrane potential $u^j(t)$ can be presented as

$$u^j(t) = \tau \cdot \sum_{i=1}^{N} \Theta(t - t_{in}^i) e^{\frac{t_d^{ij}}{\tau}} (1 - e^{-\frac{t-t_{in}^i}{\tau}}). \tag{27}$$

## A.4 ABLATION STUDY

In this section, an ablation experiment is conducted to verify the effectiveness of dendritic and axon delays. We compared the original DSNN with two modified versions containing only dendritic delay or axon delay on the four benchmark datasets. The experimental results have been presented in Table 4. We can find that both modified DSNNs perform worse than the original one. And the DSNN with only the dendritic delay is much better than the version with only the axon delay, implying the importance of dendritic delay in the DSNN. This is because the dendritic delay is functionally analogous to the synaptic weights.

Table 4: Ablation study of dendritic and axon delays.

| Tasks | dendritic delay (%) | axon delay (%) | dendritic and axon delays (%) |
|---|---|---|---|
| XOR | 99.7 | 21.4 | 100.0 |
| Iris | 95.3 | 51.3 | 96.7 |
| MNIST | 93.5 | 10.2 | 96.6 |
| FASHIONMNIST | 80.6 | 10.0 | 83.1 |

A.5    PSEUDOCODE OF THE FORWARD PROCESS OF DSNN

---

**Algorithm 2:** Forward process of one layer in the DSNN

---

**Input:** Vector of input spikes $z_{in} = e^{t_{in}}$, number of neurons in the previous and current layers $I$ and $J$, vector of activation threshold $\boldsymbol{\theta}$, matrice of dendrite delays $z_d[I][J]$, vector of axon delays $z_a[I]$;

**Output:** Vector of input spikes of the next layer's neurons $z'_{in}$;

1  $V_{id} \leftarrow \text{argsort}(z_{in})$;          // the sort indices in the ascending order
2  $\bar{z}_{in} \leftarrow z_{in}[V_{id}]$;          // sort the input spikes
3  $\bar{z}_d \leftarrow z_d[V_{id}][\cdot]$;          // rearrange the dendrite delay matrice to match the input spikes
4  **for** $j = 1$ *to* $J$ **do**
5      **for** $i = 1$ *to* $I$ **do**
6          **if** $i == I$ **then**
7              $z_{next} \leftarrow \infty$;
8          **else**
9              $z_{next} \leftarrow \bar{z}_{in}[i+1]$;
10         **end**
11         **if** $\sum_{k=1}^{i} \bar{z}_d[k,j] > \boldsymbol{\theta}$ *and* $\frac{\sum_{k=1}^{i} \bar{z}_d[k,j]\bar{z}_{in}[k]}{\sum_{k=1}^{i} \bar{z}_d[k,j] - \boldsymbol{\theta}[j]} < z_{next}$ **then**
12             $C[j] \leftarrow \{V_{si}[1], \cdots, V_{si}[i]\}$;          // causal set of the neuron
13         **end**
14     **end**
15     **if** $C[j] \neq \phi$ **then**
16         $z_{out}[j] \leftarrow \frac{\sum_{k \in C[j]} z_d[k,j]z_{in}[k]}{\sum_{k \in C[i]} z_d[k,j] - \boldsymbol{\theta}[j]}$;
17         $z'_{in}[j] \leftarrow z_{out}[j] \cdot z_a[j]$;
18     **else**
19         $z'_{in}[j] \leftarrow \infty$;
20     **end**
21 **end**
22 **return** $z'_{in}$.

---