# OpenReview forum: "Delayed Spiking Neural Network and Exponential Time Dependent Plasticity Algorithm"
_ICLR.cc/2024/Conference — ICLR 2024 Conference Withdrawn Submission_

### Official Review · Reviewer_aK4j · 2023-10-30

**Soundness:** 2 fair
**Presentation:** 2 fair
**Contribution:** 1 poor
**Rating:** 3
**Confidence:** 4

**Summary:**

This paper proposes a spiking neural network model with only synaptic and axonal delays as variables, and derives an exponential time dependent plasticity learning algorithm. Experiments on several small datasets verify the effectiveness of the method.

**Strengths:**

This paper considers the important role of synaptic/axonal delays, which is an significant feature of SNNs that is often neglected.

**Weaknesses:**

1. This paper highlights the biological plausibility, but models with only synaptic and axonal delays as variables are not biologically realistic at all. There are excitatory and inhibitory synapses and the E-I balance is one of the most important features in biological neurons. This paper totally abandons weights and takes them as 1, which is far from reality. While temporal delay is an important feature for SNNs, there is no reason to abandon weights.

2. This paper claims that the proposed ETDP is biologically plausible. Is there any reference, e.g., evidence in neuroscience, supporting such claim? As far as I know, we have no detailed evidence on how biological neurons learn delays as models for learning weights such as STDP. There are also previous methods to learn delays and the comparison between the proposed method and these methods is not discussed.

3. In the paper, it is claimed that “these foundations can be easily generalized to the DSNN with multiple hidden layers by simply updating the forward and reverse timing terms”. However, this seems highly non-trivial. For multi-layer networks with gradient-based supervised learning, it usually involves backpropagation across layers for credit assignment, which is often considered biologically implausible. It is unclear how the proposed method can solve such problem.

4. The experiments are toy and results are relatively poor. The proposed model and algorithm do not reach better STDP results. It is unclear what is the advantage of the proposed method.

**Questions:**

See weakness.

---

> ### Author Response · Authors · 2023-11-20
>
> 1. The synaptic weight is an abstract concept that measure the connection strength between neurons, which comes from ANNs. It is indeed necessary in any SNNs. The motivation of this study is not to abandon "weights" (that's a bad expression we would correct in our revised paper). We proposed our temporal-coded SNN based on the belief that the regulation of synaptic weights can be implemented by the synapses’ effects on the input spikes. Consider different lengths and shapes of axons and dendrites, and different release and absorption rates of chemical transmitters, it may be reasonable to model them as time delays added to the input spikes. In Eq. (15), we have demonstrated that the exponential term of dendritic delays in our model is functionally analogous to the synaptic weights used in conventional IF neurons. In addition, the balance of excitatory and inhibitory synapses is indeed important in biological neurons. Currently, very few SNNs can successfully use the balance to improve their performance and biological plausibility. It is a very interesting orientation; we attempt to conduct the related research in our future works.
>
> 2. We have provided abundant biological evidence to verify the biological plausibility of our proposed neural network and learning algorithm in Section 5.
> First, the proposed neuron network DSNN can be considered novel and biologically plausible. Delay is an inherent property of real biological systems. Recognizing the indispensability of spiking time delays, various studies have sought to incorporate them into SNNs. Spiking time delays are typically treated as additional learnable parameters alongside synaptic weights, resulting in increased complexity and longer running times. In contrast, our proposed DSNN uniquely employs only dendritic and axon delays as learnable parameters. To the best of our knowledge, our DSNN is the first neuron network that exclusively relies on dendritic and axon delays as learnable parameters. The comparison between our proposed method and previous delay learning approaches has been discussed in Section 5.
> Second, the proposed learning algorithm ETDP can be considered novel and biologically plausible. The ETDP algorithm is originally inspired by the gradient-based approach, but intriguingly bears a striking resemblance to the biologically plausible learning rule outlined in [1]. We compared these update strategies of two learning algorithms, and they are very similar to each other. The realizations of each term in the update strategy have been analyzed and discussed. This observation strongly supports the assertion that ETDP can be considered a biologically plausible learning algorithm. Section 5 provides the biological plausibility analysis of the DSNN and ETDP.
> [1] Roelfsema, Pieter R., and Anthony Holtmaat. "Control of synaptic plasticity in deep cortical networks." Nature Reviews Neuroscience 19.3 (2018): 166-180.
>
> 3. The ETDP algorithm does not need the accurate values of gradients to update the parameters; it instead uses approximate values for simplification. According to Eq. (16), we can find that there are three terms in the update strategy of ETDP. The term Zo-ZT can be regarded as the global signal that influences all the neurons in the network. The term Zout-Zin can be considered as the local STDP signal. Both terms determine whether to strengthen or weaken the delays. And the change amplitude is related to the forward and backward propagation time of signals. Adopting these three terms do not result in credit assignment problem. That is why the ETDP can be regarded as a biologically plausible learning algorithm.
>
> 4. The datasets used in the study are indeed not complex, but they are very commonly used to verify the performance of SNNs. Currently, SNNs cannot solve complex problems, because they use fully connected architectures and biologically plausible learning algorithms. If they use the convolution or transformer architectures and the gradient-based algorithms, their performance will become better, similar to the deep learning methods. But they will lose the biological plausibility simultaneously.

---

### Official Review · Reviewer_KQb6 · 2023-10-31

**Soundness:** 2 fair
**Presentation:** 3 good
**Contribution:** 2 fair
**Rating:** 3
**Confidence:** 4

**Summary:**

The paper introduces a Delayed Spiking Neural Network (DSNN), focusing on dendrite and axon delays as the primary learnable parameters. These delays have been observed in natural brain systems, showing their direct impact on spike timings. The DSNN is trained using a novel Exponential Time-Dependent Plasticity (ETDP) algorithm that adjusts delays based on certain biologically-inspired parameters, specifically time differences between different types of spikes and signal propagation times.

**Strengths:**

The paper is interesting as the proposed method removes the need for the resource-intensive gradient descent calculations traditionally used in ANNs and some SNNs. Experimental findings indicate that the DSNN trained using ETDP not only performs comparably to other SNN models but also maintains a high degree of biological accuracy.

**Weaknesses:**

There are certain key weaknesses of the paper: The main contribution of the paper seems to be proposing a biologically plausible SNN method.

Also, the authors point out in the introduction that "The main obstacle to SNNs is the lack of effective learning
algorithms." It seems the motivation of this work was to create a more efficient learning method with which we can achive better performance than simple ANN-to-SNN conversion. However, the performance of this proposed method seems to be worse off than current LIF SNNs with STDP learning. It would be interesting if the authors could give some more motivation and intuition of using such delayed SNNs and ETDP and why that will be better than current methods

Finally the experiments done in the paper are shown in simple datasets (XOR, Iris), and the proposed method fails to outperform current methods on these simple datasets too. This again raises the question of why we need this method in the first place.

**Questions:**

1. As stated before, it would be good if the authors could give some more motivation as to why this work is important other than biological plausibility as the notion of dealy dependent SNNs itself is not novel [1-3]

2. The authors want to make the model more biologically plausible, but it seems they are using gradient descent to learn the dendrite and the axon delay, which defeats the entire point of unsupervised learning using ETDP/STDP

3. The authors use dendrite and the axon delay in the SNN - it would be good to show the ablation study of the role each of these two delays independently play.

4. It would be interesting if the authors could add a discussion on what new modes of computation/ advantages this proposed method brings to the table.

5. I know it's difficult, but it would make the paper much more stronger if you could show the results on some datasets which leverages this new delayed SNN/ETDP model where the current models fails.








[1] Hammouamri, I., Khalfaoui-Hassani, I. and Masquelier, T., 2023. Learning delays in spiking neural networks using dilated convolutions with learnable spacings. arXiv preprint arXiv:2306.17670.
[2] Sun, P., Zhu, L. and Botteldooren, D., 2022, May. Axonal delay as a short-term memory for feed forward deep spiking neural networks. In ICASSP 2022-2022 IEEE International Conference on Acoustics, Speech and Signal Processing (ICASSP) (pp. 8932-8936). IEEE.
[3] Pham, D.T., Packianather, M.S. and Charles, E.Y.A., 2007, June. A self-organising spiking neural network trained using delay adaptation. In 2007 IEEE International Symposium on Industrial Electronics (pp. 3441-3446). IEEE.

---

> ### Author Response · Authors · 2023-11-20
>
> About the Weaknesses:
> First, the proposed neuron network DSNN is novel and biologically plausible. Delay is an inherent property of real biological systems. Recognizing the indispensability of spiking time delays, various studies have sought to incorporate them into SNNs. Spiking time delays are typically treated as additional learnable parameters alongside synaptic weights, resulting in increased complexity and longer running times. In contrast, our proposed DSNN uniquely employs only dendritic and axon delays as learnable parameters. To the best of our knowledge, our DSNN is the first neuron network that exclusively relies on dendritic and axon delays as learnable parameters. In addition, we have presented that the exponential term of dendritic delays in our model is functionally analogous to the synaptic weight used in conventional IF neurons.
>
> Second, the proposed learning algorithm ETDP is novel and biologically plausible. The ETDP algorithm is originally inspired by the gradient-based approach, but intriguingly bears a striking resemblance to the biologically plausible learning rule outlined in [1]. We compared these update strategies of two learning algorithms; they are very similar to each other. The realizations of each term in the update strategy have been analyzed and discussed. This observation strongly supports the assertion that ETDP can be considered a biologically plausible learning algorithm. Section 5 provides the biological plausibility analysis of the DSNN and ETDP.
> [1] Roelfsema, Pieter R., and Anthony Holtmaat. "Control of synaptic plasticity in deep cortical networks." Nature Reviews Neuroscience 19.3 (2018): 166-180.
>
> Third, the used benchmark problems are indeed not complex, but they are very commonly used to verify the performance of SNNs. Currently, SNNs cannot solve complex problems, because they use fully connected architectures and biologically plausible learning algorithms. If they use the convolution or transformer architectures and the gradient-based algorithms, their performance will become better, similar to the deep learning methods. But they will lose the biological plausibility simultaneously.
>
> Answers to the questions:
> 1. The biological plausibility analysis of the DSNN and ETDP has been presented in Section 5
> 2. The ETDP algorithm was originally inspired by the gradient-based approach, but it has been simplified and modified. Each term in the update strategy has been given specific biological realizations.
> 3. The ablation study of dendrite and the axon delay in the SNN has been conducted. The results have been presented in Table 4.
> 4. Since the FashionMNIST dataset is more complex than the MNIST dataset, few SNNs trained by the STDP algorithm are tested on this dataset. That led to a lack of contrasting experimental results on this problem for us. However, we have tried our best to compare our model with the other SNNs. The results have been presented in Table 3.

---

### Official Review · Reviewer_5rdv · 2023-11-01

**Soundness:** 2 fair
**Presentation:** 3 good
**Contribution:** 2 fair
**Rating:** 5
**Confidence:** 3

**Summary:**

The study presents a Delayed Spiking Neural Network (DSNN) that focuses on dendrite and axon delays as the primary learnable parameters, emphasizing biological plausibility. To train this DSNN, an Exponential Time-Dependent Plasticity (ETDP) algorithm is introduced. This algorithm adjusts delays based on time differences between presynaptic and postsynaptic spikes and signal propagation times. The DSNN, when trained with ETDP, achieves competitive performance on benchmark datasets while retaining a high degree of biological authenticity.

**Strengths:**

The paper's contributions lie in the novel DSNN design, the biologically plausible ETDP training algorithm, and the demonstrated performance on machine learning tasks.

**Weaknesses:**

1.	While the authors have proposed ETDP for the training of SNNs, based on Eq.7, this method seems to assume \tau=1, implying that it is only applicable to IF neurons. However, the IF neuron is a highly simplified neuron model, which does not align well with the biological plausibility that the authors claim for this method.
2.	Although the authors claim that this approach can easily be extended to deep architectures of DSNN with multiple hidden layers, there is no experimental evidence to support this.
3.	The experiments are quite basic, and the results are not very convincing. The authors have only conducted experiments on the XOR problem, IRIS dataset, and the MNIST/FashionMNIST datasets. For the FashionMNIST dataset, only the accuracy is reported in the paper, without any other benchmark results for comparison. An accuracy of 96.6% on the MNIST dataset is not very compelling, especially when, as the authors mention in Tab.2, previous works have achieved better accuracy than ETDP. Additionally, under the same experimental settings, ETDP's performance is also inferior to GD.

**Questions:**

1.	In Eq.1, what is \kappa?
2.	In Fig.1, although I can gather information about this figure from the related description in Section 3, the figure does not intuitively convey to readers what its elements represent, especially the relationship between i-1, i+1, and i. Are they neurons from different layers or something else?
3.	I understand the relationship between Section 3 and Section 5, but when Section 4 discusses the non-linearity of spiking neurons, I'm not clear on how this relates to the main theme of the paper.
4.	In Tab.1 and Tab.2, the authors should highlight the method with the best performance, not their own. Highlighting their method can mislead the readers. Also, I suggest separating the results of the proposed method from other methods with horizontal lines.
5.	Given that gradient information is used and BP performance surpasses ETDP, what are the distinctions and advantages of ETDP over BP?
6.	For the XOR problem and the results on FashionMNIST, I suggest that the authors provide more analysis and supporting materials to increase the reader's confidence. This will also offer a more intuitive understanding of the method's performance, rather than just briefly mentioning accuracy in the paper.

---

> ### Author Response · Authors · 2023-11-20
>
> About the Weaknesses:
> 1. Indeed, the IF neuron is a highly simplified neuron model. It is s a simple one-dimensional model of a biological neuron that needs less computational effort for modelling a biological neuron. The Hodgkin and Huxley model is a four-dimensional model that can simulate the dynamic of a biological neuron with more details. But it has a very high computational cost. Most SNNs used the IF neurons in their structures. \tau is a constant hyper-parameter, and it is common to set \tau=1 to simplify the calculation of SNNs. The biological plausibility of the proposed method has been analyzed in Section 5. We provided a great amount of biological evidence to verify its biological plausibility.
> 2. In theory, the proposed approach can be extended to deep architectures of DSNN with multiple hidden layers. It is because, first, the proposed learning algorithm ETDP can be extended to multiple hidden layers, and the update strategies are the same for all the layers. Second, the activation function is proven non-linear in the DSNN. But, we have to admit that, currently, the SNNs cannot solve very complex problems such as the Imagenet dataset, if they use fully connected architectures and biologically plausible learning algorithms. The benchmark problems used in this study are too simple; SNNs do not need deep architectures to solve them.
> 3. As mentioned above, the used benchmark problems are not complex, but they are very commonly used in the field of SNNs. Even most SNNs are not verified on the FashionMNIST dataset. They cannot solve complex problems because they use fully connected architectures and biologically plausible learning algorithms. If they use the convolution or transformer architectures and BP algorithms, their performance will be better, as the deep learning methods. But they will lose the biological plausibility in this way. Our main contributions of this study can be summarized as follows. First, a novel DSNN is proposed solely with dendritic and axon delays, without the commonly used synaptic weights, which can yield competitive performance. To the best of our knowledge, our DSNN is the first neuron network that exclusively relies on dendritic and axon delays as learnable parameters. Second, the proposed learning algorithm ETDP has a high degree of biological plausibility. It intriguingly bears a striking resemblance to the biologically plausible learning rule outlined in [2, 3].
> [2] Roelfsema, Pieter R., and Anthony Holtmaat. "Control of synaptic plasticity in deep cortical networks." Nature Reviews Neuroscience 19.3 (2018): 166-180.
> [3] Fisher, Simon D., et al. "Reinforcement determines the timing dependence of corticostriatal synaptic plasticity in vivo." Nature communications 8.1 (2017): 334.
>
> Answers to the questions:
> 1. \kappa is the kernel function of IF neurons. We have added its description to the revised manuscript.
> 2. i, j and k are the indexes of neurons on the input, hidden and output layers, respectively. We have added the corresponding description.
> 3. The non-linearity of activation function is an important property of extending the SNNs to deep architectures, similar to artificial neural networks.  Thus, we discuss the non-linearity of spiking neurons in Section 4. However, due to the page limit, the part has been placed into the Appendix.
> 4. According to your suggestion, the best methods have been highlighted, and the results of the proposed method have been separated from other methods with horizontal lines.
> 5. The BP algorithm uses the gradient information directly; thus it has better optimization performance. But it is thought to be biologically unrealistic. In converse, our proposed ETDP, originally inspired by the gradient-based approach, has been verified to obtain a high degree of biological plausibility. The biological evidence has been added to Section 5. The ETDP uses the approximate values of gradients. That is why it performs worse than the BP algorithm. In fact, the phenomenon is not only observed on our DSNN, but also on other SNNs, such as [1].
> [1] Payeur, Alexandre, et al. "Burst-dependent synaptic plasticity can coordinate learning in hierarchical circuits." Nature neuroscience 24.7 (2021): 1010-1019.
> 6. The XOR problem is the simplest nonlinear problem. Most SNNs can perform well on this problem. We used it to verify the effectiveness of our proposed method on this nonlinear problem. However, the FashionMNIST dataset is more complex than the MNIST dataset. Few SNNs trained by the STDP algorithm are tested on this dataset. That led to a lack of contrasting experimental results on this problem. But we have tried our best to compare our model with the other SNNs. The results have been presented in Table 3.